# Intradermal injection of lidocaine with a microneedle device to provide rapid local anaesthesia for peripheral intravenous cannulation: A randomised open-label placebo-controlled clinical trial

**Alexey Rzhevskiy**[1]*, **Andrei Popov**[1], **Chavdar Pavlov**[2], **Yuri Anissimov**[3], **Andrei Zvyagin**[1], **Yotam Levin**[4], **Efrat Kochba**[4]

**1** Center of Biomedical Engineering, Sechenov First Moscow State Medical University, Moscow, Russia,
**2** Clinic of Internal Diseases Propedeutics, Sechenov First Moscow State Medical University, Moscow,
Russian Federation, **3** School of Natural Sciences, Griffith University, Gold Coast, Queensland, Australia,
**4** NanoPass Technologies, Nes Ziona, Israel

\* rzhevskiy01@gmail.com

**Data Availability Statement:** All relevant data are within the manuscript and its Supporting

## Abstract

### Background

Peripheral venous cannulation is one of the most common procedures in medicine. It is associated with noticeable pain and apprehension, although in most cases it is performed without any anesthesia due to lack of a painless, cost-effective option, which would provide rapid local anesthesia with subsequent significant reduction in the experienced pain. We conducted an open-label placebo-controlled clinical trial to evaluate the efficacy and safety of a 2% lidocaine injection using the commercially available microneedle device Minron-Jet600 (NanoPass Technologies Ltd, Israel) to achieve rapid local anesthesia prior to peripheral venous cannulation.

### Methods

One hundred and two subjects were randomly allocated into two groups. In the first group, 100μL of lidocaine hydrochloride (2%) was injected intradermally to subjects using the MicronJet600 device in the left arm (MJ-Lido) and 100μL of saline was injected intradermally using the device in the right arm (MJ-Saline). In the second group, 100μL of lidocaine hydrochloride (2%) was injected using the MicronJet600 device into the left arm (MJ-Lido), with no injection into the right arm of subjects (No pretreatment). In both groups the intradermal injection was performed at the cannulation site prior to insertion of a 18G cannula into a median cubital vein in both arms. As a primary variable, a score of cannulation-induced pain was indicated by subjects using a 100-point visual analog scale immediately after cannulation. As a secondary variable, subjects in Group 2 also indicated their preference to receive the anaesthetic injection with MicronJet600 in the future by using the 5-point Likert scale. Also, as a secondary variable, the duration of skin numbness after lidocaine injection was

Information files, and public repository
ResearchRegistry: (registration number:
researchregistry4662, principal investigator:
Chavdar Pavlov, date of registration: 29 January
2019, URL: https://www.researchregistry.com/
browse-the-registry#home/registrationdetails/
5c4d811ac413740862094f0f/).

**Funding:** The work was supported by the Ministry
of Education and Science of the Russian
Federation, grant No. 075-15-2019-192. NanoPass
Technologies Ltd. provided support in the form of
salary for authors EK and YL. A payment for labour
of the nurse who participated in the study was
covered by NanoPass Technologies Ltd. as well.
The funders had no role in study design, data
collection and analysis, decision to publish, or
preparation of the manuscript. The specific roles of
the authors are articulated in the 'author
contributions' section.

**Competing interests:** The authors have read the
journal's policy and have the following competing
interests: EK and YL are paid employees of
NanoPass Technologies Ltd. the developer of the
MicronJet600 device. This does not alter our
adherence to PLOS ONE policies on sharing data
and materials. There are no patents, products in
development or marketed products associated with
this research to declare.

indicated by performing a superficial pin-prick with a 27G needle at 15, 30 and 45 minutes, at distances of 1, 2 and 3 centimeters from the injection site.

## Results

A significant pain reduction (11.0-fold) was achieved due to the lidocaine injection compared to the cannulation without any pretreatment ($p < 0.005$). After the lidocaine injection the anesthesia was effective up to 2 centimeters from the injection site and remained for up to 30 minutes. Eighty percent of subjects from the second group preferred cannulation after the lidocaine injection over cannulation without any pretreatment. No significant side effects were identified.

## Conclusion

Intradermal injection of anaesthetic with Micronjet600 was found to be a safe and effective option for providing rapid local anesthesia for peripheral intravenous cannulation.

## Trial regiatration

The clinical trial was registered, before the patient enrollment began, in the Research Registry publicly accessible database (registration identifier: researchregistry4662). Also, the trial was registered in ClinicalTrials.gov (registration identifier: NCT05108714) after its completion.

## Introduction

Intravenous cannulation is a common painful procedure which is, however, usually performed without local anaesthesia [1]. The simplest approach involving injection of a local anaesthetic into the skin using a regular needle is, in itself, painful therefore several techniques were previously tested for reducing pain in intravenous cannulation, with each having specific limitations which reduce convenience [2–7]. Intravenous cannulation requires local anaesthesia, which simultaneously provides an immediate effect, cost-effectiveness and simplicity, with a minimum of discomfort to the patient [8].

The use of hollow microneedles is currently one of the most promising techniques for providing local anaesthesia in superficial interventions involving skin and subcutaneous adipose tissue, in particular for peripheral venous cannulation [9]. To date, several commercially available, microneedle-based devices can be found on the market. Among them is the hollow microneedles based device, MicronJet600 (MJ600) by NanoPass Technologies Ltd, Israel, which was approved by regulatory authorities in many territories, including the United States and the European Union. MicronJet600 was primarily investigated as a device for nearly-painless [10] intradermal injection of vaccines [10–18]. The device is also considered promising for use in other intradermal applications [19], including intradermal injection of anaesthetics.

To test the efficacy of MicronJet600 to provide rapid local anaesthesia for peripheral intravenous cannulation via intradermal injection of micro-amounts of anaesthetic, with a subsequent decrease of the intervention-related pain score as a primary variable, an open-label placebo-controlled clinical trial was conducted. To assess safety of the intervention, potential side effects were estimated. Further, preference of cannulation, preceded by the intradermal

injection of anaesthetic, over the cannulation without any pretreatment, duration and area of skin numbness after the lidocaine injection, were assessed as the secondary variables.

## Materials and methods

The trial was registered prior to patient enrollment in the Research Registry publicly accessible database (registration identifier: researchregistry4662, principal investigator: Chavdar Pavlov, date of registration: 29 January 2019, URL: https://www.researchregistry.com/browse-the-registry#home/registrationdetails/5c4d811ac413740862094f0f/). Also, the trail was registered in ClinicalTrials.gov (registration identifier: NCT05108714) after its completion. The authors confirm that all ongoing and related trials for this drug/intervention are registered. The study received ethical approval from the Local Ethics Committee of First Moscow State Medical University (Extract from Minutes No. 07–17 of the Local Ethics Committee meeting of 13.09.2017) and written informed consent was obtained from all subjects participating in the trial. The study was conducted in accordance with the Declaration of Helsinki (2013) protocol and CONSORT (Consolidated Standards of Reporting Trials) (Fig 1). Study subjects, healthy volunteers and patients at University's Clinical Hospital 2 (Moscow, Russia) were enrolled between January 29th, 2019 and March 15th, 2019. The recruitment ended after the number of enrolled participants exceeded the designated sample size for each group.

### Study design

A single center, open-label placebo-controlled clinical trial to evaluate the efficacy and safety of 2% lidocaine injection, using the commercially available microneedle device MinronJet600 (NanoPass Technologies Ltd, Israel), to achieve rapid local anesthesia prior to peripheral venous cannulation.

### Study objectives

The primary objective was to evaluate, in terms of VAS score, the efficacy of intradermal administration of low doses of lidocaine 2% solution using MicronJet600, to reduce the pain associated with peripheral venous catheter insertion. Secondary objectives included identification of potential side effects from intradermal administration of lidocaine with the

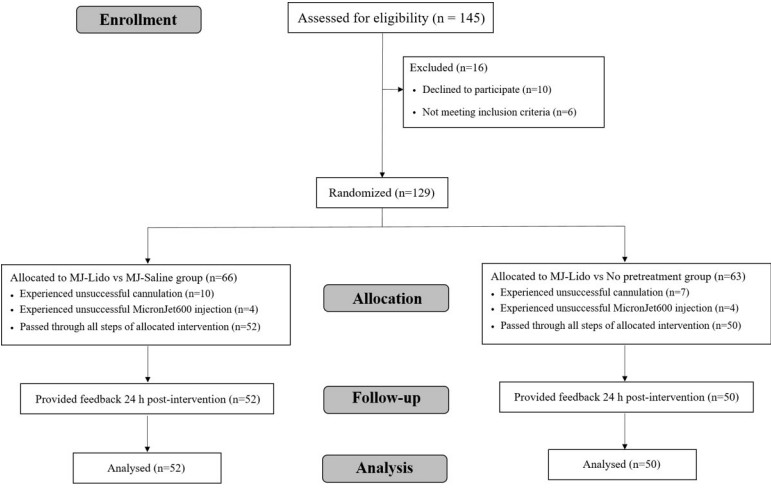

**Fig 1. Consolidated standards for reporting of trials diagram.**

MicronJet600 and assessment of the area and duration of skin numbness by performing a gentle superficial pinprick with 27 G hypodermic disposable needle, at various distances from the injection site, at various time points. Subjects' preference for local anaesthetic injection with MicronJet600 prior to future cannulations was also assessed.

## Participants

One hundred and two healthy volunteers were pre-screened for eligibility. Inclusion criteria included any gender, age between 18–65 years, and absence of all exclusion criteria. The main exclusion criteria included pregnancy or breast feeding, evidence of allergy to lidocaine, presence of pain of any localization and character not associated with the study, or treatment with any analgesics, any local tissue damage at the site of intervention, and serious systemic diseases. After being considered eligible for the study and signing the informed consent form, the subjects were randomly allocated into two groups by the first observer AR (Observer1). Simple randomization was performed to allocate subjects into two groups using the Microsoft Excel random number generator. The subjects who were allocated random even numbers were assigned to the first study group (MJ-Lido vs MJ-Saline) while the subjects who were allocated random odd numbers were assigned to the second study group (MJ-Lido vs No pretreatment).

## Intervention

Prior to the intervention, each subject had his or her median cubital vein identified by palpitation of the cubital fossa area by a nurse, to determine the site for intravenous cannulation. Further, the cannulation site was wiped with ethanol swabs. Each subject from the MJ-Lido vs MJ-Saline group received an injection of 100 µL of 2% lidocaine hydrochloride injectable solution (Biokhimik, Russia) into the left arm at the cannulation site and an injection of 100 µL of saline solution (Biokhimik, Russia) placebo into the right arm at the corresponding site. Each injection was immediately (t = 0) followed by cannulation with an 18 G peripheral venous catheter. In the second group MJ-Lido vs No pretreatment, each subject received the injection of 100 µL of 2% lidocaine into the left arm at the cannulation site which was followed by the cannulation with an 18 G catheter, while the right arm of each subject was cannulated with an 18 G catheter without any pretreatment. Thus, each subject was his or her own control. This trial design was chosen to identify the presence or absence of a placebo-related effect.

The injections of both lidocaine and placebo were performed with MicronJet600 (Fig 2(A)) placed on a 1 mL syringe (Fig 2(B)), prefilled with a 27G needle. The injection procedure lasted approximately 4 seconds with the a flow rate of approximately 25 µl/sec. The intradermal

(A)                                          (B)

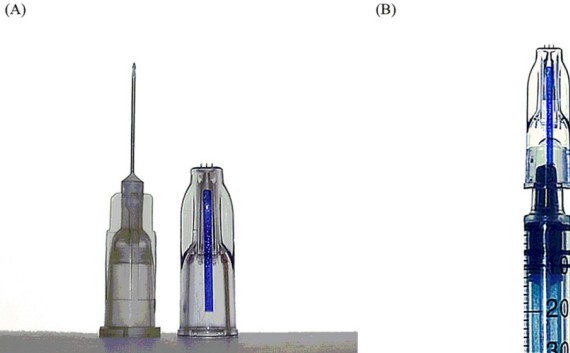

**Fig 2.** MicronJet600 compared to the 27 G hypodermic needle (A), and MicronJet600 placed on 1 mL syringe (B).

injection with MicronJet600 was considered successful if a bleb, of approximately 10–15 mm in width and 3–6 mm in height, was formed at the site of injection. A cannulation was considered successful when a small amount of blood was present in the cannula's hub following the cannula insertion. Each cannulation was performed by moving the cannula only forward when inserting into the vein. In case of unsuccessful insertion, any attempt at reinsertion was prohibited. Immediately after the insertion, the cannula was removed and the site of cannula insertion was then wiped with ethanol swabs and covered with an adhesive bandage. All injections and cannulations were performed by the same highly-qualified staff nurse of the University's Clinical Hospital 2, who had previously undergone training on the proper use of MicronJet600 based on the training materials provided by NanoPass Technologies Ltd, Israel.

After each cannulation, as a primary endpoint variable, the subjects scored the pain experienced using a 100-point visual analog scale (VAS), ranging from no pain (0) to unbearable pain (100) [20], presented by the second observer AP (Observer2), and the scores were recorded. The pain experienced by subjects due to cannulation in each of the cases (following lidocaine injection, placebo injection, or without pretreatment) was also evaluated in terms of the VAS-score. Thus, in the context of the current study, VAS-score = 0 was considered as a lack of pain, VAS-score≤10 as a mild pain score, VAS-score≤20 as an acceptable pain score, VAS-score>20 as an unacceptable pain score. s a secondary endpoint variable, the duration of skin numbness due to lidocaine injection was assessed by performing a gentle superficial pinprick with a 27 G hypodermic disposable needle, perpendicularly to the arm at the distance of 1, 2 and 3 cm from the injection site in the distal direction at 15 (t = 15), 30 (t = 30) and 45 (t = 45) minutes after the injection. The pinpricks were performed by the Observer1. For each subject, a single 27 G hypodermic needle was used at each time point, and the needle disposed of after the procedure. The pain experienced due to the pinpricks was also assessed, by the subjects, in accordance with the provided 100-point VAS scale and recorded by Observer2. After the cannulations were performed in both arms of the subjects in the MJ-Lido vs No pretreatment group, the subjects were asked whether they would prefer to receive an anaesthetic injection with MicronJet600 prior to cannulations in future. After the cannulations were performed in both arms of the Group 1 subjects (MJ-Lido vs MJ-Saline), the subjects were asked whether they would prefer to receive anaesthetic injection with MicronJet600 prior to the cannulations in the future. The preference assessment was performed with the 5-point Likert scale where 1 was defined as strong disagreement, 2 as disagreement, 3 as lack of any preference, 4 as agreement and 5 as strong agreement. To assess possible side effects of the intervention, the cannulation site was examined for evidence of swelling, edema, hematoma, or hemorrhage at 60 minutes after the procedure. Further, the subjects were contacted by phone, 24 hours after the injection, and asked about any evidence of study-related adverse events. A general study scheme is presented in Fig 3.

## Statistical analysis

Regression modeling and results visualization were performed using R (version 3.6.3) environment for statistical computing (R Foundation for Statistical Computing, Vienna, Austria) and third-party packages lme 4 1.1–21, clubSandwich 0.4.1 and emmeans 1.4.8 available on the CRAN repository. Linear mixed effects models (implemented in the lme4 1.1–21 package) were used to model VAS scores after interventions: assuming random intercepts, random slope for repeated measurements (corresponding to coefficients for 30 min and 45 min) for each study participant, and measurement time (15, 30 and 45 min)–distance (1, 2 and 3 cm) interaction. For all models Sandwich cluster-robust variance-covariance matrix estimators (implemented in the clubSandwich 0.4.1 package) were used to address heteroskedasticity, the

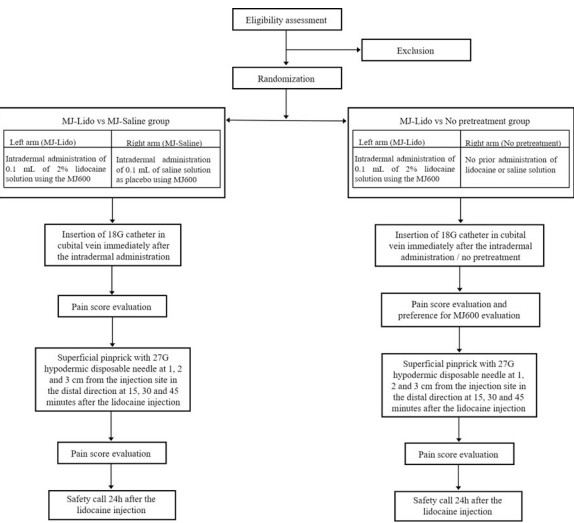

**Fig 3. General study scheme.**

Satterthwaite method was used to approximate degrees of freedom and the Tukey method was used to adjust p-values obtained from pairwise comparisons. Cohen's d was used as a standardized effect size estimate.

## Sample size

A sample size of 40 subjects per group was calculated to detect an effect size of (expected difference on the VAS score between two time points at a specific distance) 1 with standard deviation in the effect of 2.2, using a paired t-test with 80% power and 5% type I error rate assuming a two-sided significance testing procedure. At the same time, an additional 22 subjects (102 subjects in total) were enrolled in order to compensate for dropouts.

## Results

One hundred and twenty-nine subjects gave informed consent and were enrolled in the study; 66 subjects were allocated into Group 1 (MJ-Lido vs MJ Saline) and 63 into Group 2 (MJ-Lido vs No pretreatment). Ten subjects from Group 1 and 7 subjects from Group 2 had at least one unsuccessful cannulation; these subjects were excluded from the study. Four subjects from the Group 1 and 6 subjects from Group 2 were also excluded from the study due to unsuccessful injections with MicronJet600 at the first attempt. In these cases, owing to deviations in the technique for the injection, insufficient penetration of the microneedles into the skin led to a major leakage of the injected solution onto the skin (10 out of 186 injections, 5.4%, resulted in major leakage). Thus, data from 52 subjects from MJ-Lido vs MJ Saline (Group 1) and 50 subjects from MJ-Lido vs No pretreatment (Group 2) were analyzed (Table 1).

The results from the linear mixed effects model of VAS score after the cannulation are presented in S1 Table. According to the results (Fig 4), the mean pain score of the cannulation was 3.6 (95% CI from 2.6 to 4.6) for the MJ-Lido, 41.5 (95% CI from 38.2 to 44.8) for the MJ-Saline and 39.7 (95% CI from 35.7 to 43.7) in the absence of pretreatment. The pain reduction effect caused by intradermal administration of 100 µL of 2% lidocaine compared with both saline injection and no pretreatment was statistically significant ($p < 0.0001$) with corresponding Cohen's d estimates -4.5 (95% CI from -4.9 to -4.2) and -4.3 (95% CI from -4.8 to

**Table 1. Baseline demographic characteristics.**

| Characteristics | Treatment group | |
| --- | --- | --- |
| | MJ-Lido vs MJ-Saline (N = 52) | MJ-Lido vs No pretreatment (N = 50) |
| Sex—no (%) | | |
| Male | 35 (67%) | 29 (58%) |
| Female | 17 (33%) | 21 (42%) |
| Age–years (±) | | |
| Min | 18 | 18 |
| Max | 59 | 63 |
| Mean | 28.6 (±11.3) | 30.2 (±13.6) |
| Median | 24.5 | 28.4 |
| Body Mass Index (BMI) | | |
| Mean (SD) | 24.8 (±3.6) | 25.3 (±3.1) |
| Range | 18.8–34.3 | 17.4–31.7 |

-3.9). Also, no placebo-related effect was determined (Cohen's d = 0.2, 95% CI from -0.4 to 0.8, p = 0.8).

The distribution of cases between four VAS-score groups (VAS-score = 0, VAS-score≤10, VAS-score≤20 and VAS-score>20) was estimated in percentage for the scenarios with the lidocaine (n = 102) or placebo injection (n = 52) prior to cannulation, or without any pretreatment (n = 50). Thus, the distribution of cases in the scenario when lidocaine injection preceded the cannulation was 54.9%, 95.1%, 100% and 0% for VAS-score = 0, VAS-score≤10,

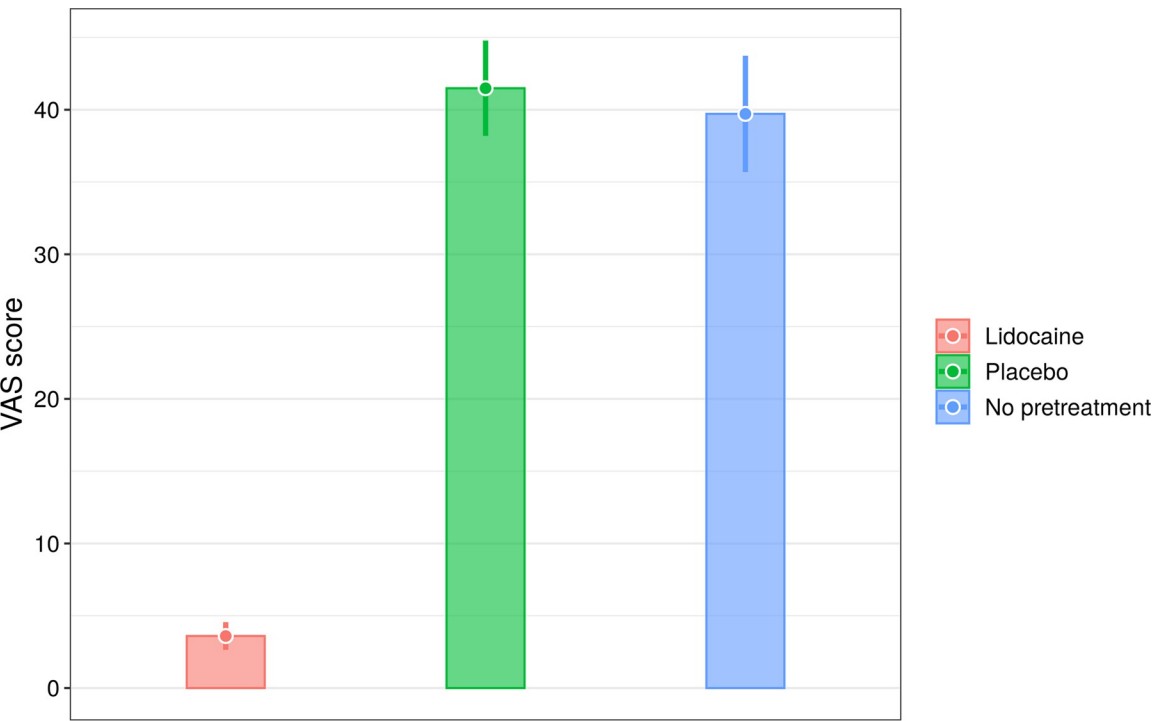

**Fig 4. Mean estimates of VAS scores with corresponding 95% confidence intervals for pain experienced by the subjects after cannulations preceded by the lidocaine injection (MJ-Lido vs MJ-Saline and MJ-Lido vs No pretreatment groups (red bar), saline injection (MJ-Lido vs MJ-Saline group, green bar), or performed without any pretreatment (MJ-Lido vs No pretreatment group, blue bar).**

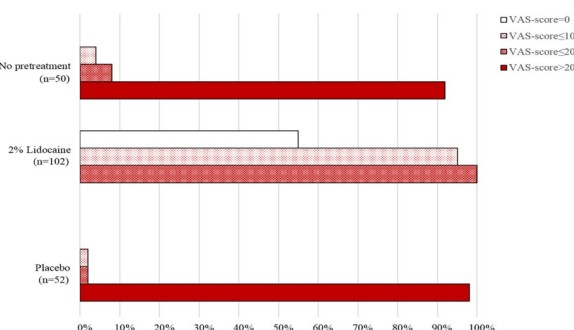

**Fig 5. The bar chart shows the distribution of subjects between the groups ranked by VAS-score for the cases of the injection of 100 µL of 2% lidocaine or placebo prior to cannulation, or cannulation without any pretreatment.** Thus, the non-shaded bar, slightly shaded bar, moderately shaded bar and entirely shaded red bar represents the percentage of subjects related to the groups: VAS-score = 0, VAS-score≤10, VAS-score≤20, VAS-score>20, respectively.

VAS-score≤20 and VAS-score>20, respectively; the distribution of cases in the scenario when placebo injection preceded the cannulation was 0%, 1.9%, 0% and 98.1% for VAS-score = 0, VAS-score≤10, VAS-score≤20 and VAS-score>20, respectively; the distribution of cases in the scenario when cannulation was performed without any pretreatment was 0%, 4%, 8% and 92% for VAS-score = 0, VAS-score≤10, VAS-score≤20 and VAS-score>20, respectively (Fig 5).

The results from the linear mixed effects model of VAS score after the cannulation are presented in S2 Table. The dependence of skin numbness after the intradermal lidocaine injection was determined to be statistically significant for both predictors: distance of the pinprick from the injection site with 27G needle, time after the injection and their interaction (p<0.0001). As expected, skin numbness was significantly higher at t = 15 and the distance of 1 cm with the mean VAS of 4.5 (95% CI from 3.5 to 5.5) as compared to other time points and distances: 9.0 (95% CI from 8.1 to 10.0), 10.9 (95% CI from 9.7 to 12.0), 11.0 (95% CI from 9.8 to 12.2), 12.4 (95% CI from 11.0 to 13.8), 12.2 (95% CI from 10.9 to 13.6), 11.7 (95% CI from 10.6 to 12.7), 12.3 (95% CI from11.0 to 13.5), 12.1 (95% CI from 11.1 to 13.2) for t = 15 and 2cm, t = 15 and 3cm, t = 30 and 1 cm, t = 30 and 2cm, t = 30 and 3cm, t = 45 and 1cm, t = 45 and 2cm, t = 45 and 3cm, respectively (Table 2). Fig 6 depicts the alteration of the average pain scores at three time points in relation to the distance from the injection site. At the end of the study (t = 60), no subjects indicated a feeling of skin numbness at the injection site.

Adverse events of lidocaine injection with MicronJet600 were visually assessed right after the injection (t = 0) and at the end of the study (t = 60min); Thus, at t = 0, there were no adverse events indicated. A bleb (wheal) of 10–15 mm in length and 3–6 mm in height was formed in all subjects immediately after the injection, which is considered a sign of successful intradermal injection. At t = 60 min, a slight erythema of the injection site was noticeable in

**Table 2. Mean estimates with corresponding 95% confidence intervals for VAS pain score due to pin-pricks with a 27G needle at three time points at 15, 30 and 45 minutes after the lidocaine injection with MicronJet600, and distances at 1, 2 and 3 centimeters from the injection site.**

| Time (min) | Distance (cm) | | |
|---|---|---|---|
| | **1** | **2** | **3** |
| **15** | 4.5 (95% CI: 3.5–5.5) | 9.0 (95% CI: 8.1–10.0) | 10.9 (95% CI: 9.7–12.0) |
| **30** | 11.0 (95% CI: 9.8–12.2) | 12.4 (95% CI: 11.0–13.8) | 12.2 (95% CI: 10.9–13.6) |
| **45** | 11.7 (95% CI: 10.6–12.7) | 12.3 (95% CI: 11.0–13.5) | 12.1 (95% CI: 11.1–13.2) |

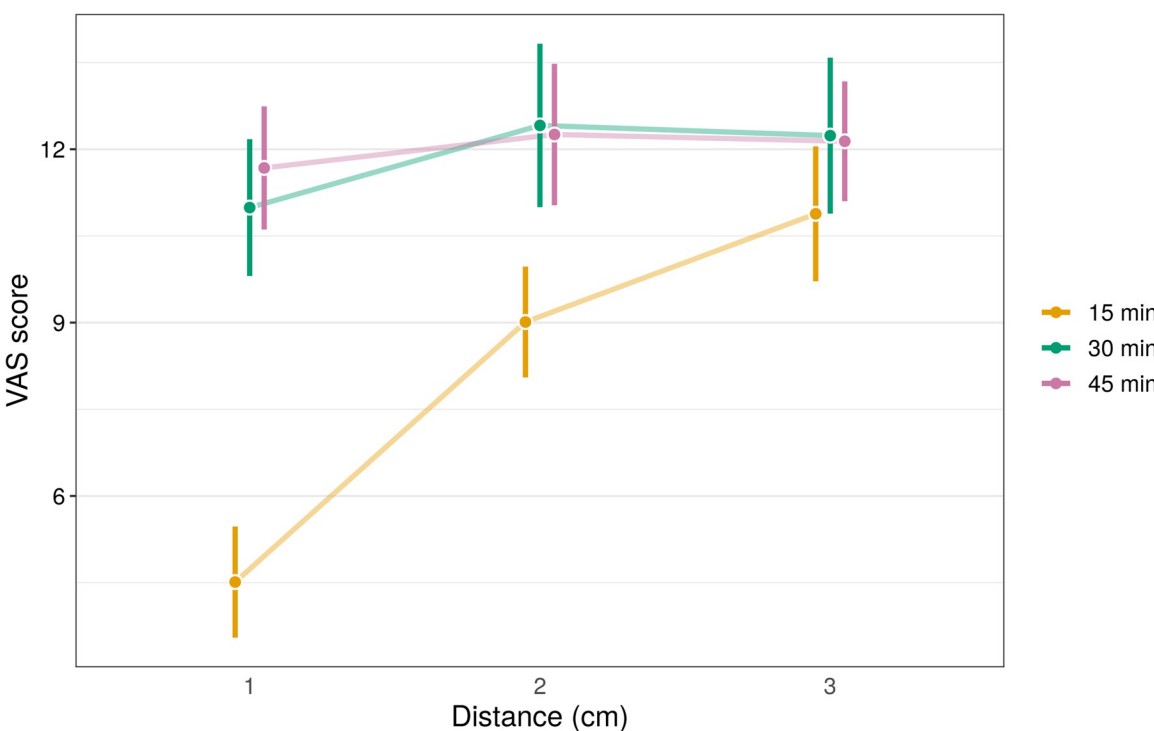

**Fig 6. The multi-line chart demonstrates mean estimates of VAS scores with 95% confidence intervals for pain experienced by subjects due to the superficial pin-pricks with 27 G needle for three different time points at 15, 30 and 45 minutes after the lidocaine injection with MicronJet600, and at 1, 2 and 3 centimeters from the injection site.**

subjects with pale skin. Further, 24 hours after the study, there were no reports of erythema or swelling at the site of lidocaine injection and cannulation. At the same time, 9 subjects (8.8% in total from Group1 and Group2), 5 subjects (9.6%) and 4 subjects (8%) complained of local hematoma around the cannulation site for the scenarios when cannulation was performed after the lidocaine injection, after placebo injection, or without any pretreatment, respectively. However, local hematoma is considered a common adverse event of the cannulation itself and there was no evidence for correlation between the injections with MicronJet600 prior to cannulation and an increased prevalence of hematomas.

## Discussion

Currently, there is an unmet need in clinical practice in which common painful procedures, including intravenous cannulation, are performed without proper or any anaesthesia. This can cause pain, anxiety and discomfort to patients. In this study, the efficacy and safety of intradermal administration of anaesthetic with MicronJet600 to provide local anaesthesia for peripheral intravenous cannulation was tested. According to the results of this open-label placebo controlled clinical trial, the intradermal injection of 100 μL of 2% lidocaine with MicronJet600 significantly decreased the pain score experienced by subjects due to insertion of 18 G cannula into a median cubital vein. The difference between pain scores experienced due to intravenous cannulation with and without local anaesthesia provided by the lidocaine injection substantially exceeded the average clinically significant difference of 9–13 on the 100-point VAS [21, 22]. Further, there were no statistically significant differences between the average VAS-scores due to cannulations after placebo injection and without any pretreatment, which demonstrates the absence of a placebo-related effect. Moreover, intradermal injection of 2% lidocaine with

MicronJet600 provided local anaesthesia for 15–30 minutes, and therefore can be effectively used in diverse cases of mid-term surgical intervention involving skin and subcutaneous fat. No significant adverse events from the intervention were identified.

At the same time, this study has several limitations such as the number of subjects per group; time points for pain score measurement after the intravenous cannulation; volume and concentration of the anaesthetic and type of anaesthetic. Also blinding could has been performed more rigorously, although it is difficult to blind (disguise) the use of the MicronJet600 device; the intraindividual comparisons are confounded with the application side, which has an unclear effect on the study result; since there is no evidence regarding the difference of pain sensitivity between arms in population, arms were not randomised. Further, it is worth mentioning that the comparison within Group2 is confounded with the MicronJet600 use. In addition, the study was not powered for adverse events. Finally, the study did not involve a control group whereby subjects would receive regular intradermal injection of lidocaine with a hypodermic needle.

Although no direct comparison was made between local anaesthesia with the MicronJet600 and its most competitive alternative, Jet injectors, it is anticipated that the use of MicronJet600 is more effective. In a clinical trial by Lysakowski, Dumont, Tramer, Tassoniy [23] the effectiveness of local anaesthesia with intradermal jet injection of lidocaine with J-Tip (National Medical Products Inc, CA, USA) was investigated; the average pain scores, experienced by subjects following a 18-G cannula insertion into a vein on the dorsal part of the arm and measured with 10-point Numerical Verbal Scale (NVS), were: 3.9, 4.2 and 1.7 for the scenarios of cannulation without pretreatment, cannulation after the injection of 500 μL of saline and cannulation after the injection of 500 μL of 2% lidocaine, respectively. Consequently, the average pain scores for cannulation with no pretreatment, and cannulation with the preliminary intradermal injection of the placebo, were comparable between the current study and the study by Lysakowski, Dumont, Tramer, Tassoniy [23]. Thus, the average 100-point VAS pain scores versus 10-point NVS were: 39.7 vs 3.9 for cannulation without any pretreatment, and 41.5 vs 4.2 for cannulation after the placebo injection. The reduction in the average pain score of the cannulation by intradermal administration of 2% lidocaine, however, was substantially higher in case of the MicronJet600 intradermal administration. It resulted in an 11.0-fold reduction (from 39.7 to 3.6) in VAS pain score, compared to the jet injection intradermal administration which resulted in only 2.3-fold reduction from 3.9 to 1.7 in NVS pain score. Moreover, in the current study, a significantly lower amount (100 μL) of 2% lidocaine was administered in comparison with the study by Lysakowski, Dumont, Tramer, Tassonyi (500 μL) [23] which illustrates further the greater effectiveness of MicronJet600 as a tool for providing intradermal administration of anaesthetics to achieve rapid local anaesthesia over the jet injection method.

The adverse events of cannula insertion after the lidocaine injection with MicronJet600 were insignificant. The only obvious sign of the injection was the formation of a bleb, which is considered a sign for successful intradermal injection. Additionally, as the intradermal injection of only a small amount (100 μL) of 2% lidocaine with MicronJet600 was sufficient to achieve the substantial reduction of pain, the technique is considered safe in terms of prevention of serious complications if the injection was accidentally performed in a subject with lidocaine hypersensitivity.

## Conclusions

Overall, intradermal administration of low doses of lidocaine 2% solution with MicronJet600 is effective in reducing the pain associated with a peripheral venous catheter insertion procedure, providing a sufficient rate of local anaesthesia immediately post-injection. No significant

adverse events were associated with the intervention, which signifies its high safety. Further, 80% of subjects from the MJ-Lido vs No pretreatment group preferred cannulation after the lidocaine injection over the cannulation without any pretreatment.

## Supporting information

**S1 Checklist.**
(DOC)

**S1 Table. The results of the linear mixed effects model of VAS score after the cannulation.**
(DOCX)

**S2 Table. The results from the linear mixed effects model of skin numbness after the lidocaine injection.**
(DOCX)

**S1 File.**
(PDF)

**S2 File.**
(PDF)

## Acknowledgments

We would like to thank clinical and administrative staff of University Clinical Hospital 2 of Sechenov University for participating in this study. Additionally, we are grateful to all volunteers participated in this study.

## Author Contributions

**Conceptualization:** Alexey Rzhevskiy, Yotam Levin, Efrat Kochba.

**Investigation:** Alexey Rzhevskiy, Chavdar Pavlov, Yuri Anissimov.

**Methodology:** Alexey Rzhevskiy, Efrat Kochba.

**Project administration:** Alexey Rzhevskiy, Chavdar Pavlov.

**Resources:** Chavdar Pavlov, Yotam Levin.

**Software:** Alexey Rzhevskiy, Yuri Anissimov.

**Supervision:** Yotam Levin, Efrat Kochba.

**Validation:** Efrat Kochba.

**Visualization:** Efrat Kochba.

**Writing – original draft:** Alexey Rzhevskiy, Andrei Popov, Chavdar Pavlov, Yuri Anissimov, Yotam Levin, Efrat Kochba.

**Writing – review & editing:** Alexey Rzhevskiy, Andrei Zvyagin, Yotam Levin, Efrat Kochba.

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
