## [Decision Letter · Decision Letter 0]

4 Feb 2020

PONE-D-19-27010

Intradermal injection of lidocaine with a microneedle device to provide rapid local anaesthesia for peripheral intravenous cannulation: A randomised double-blind placebo-controlled clinical trial

PLOS ONE

Dear Dr Rzhevskiy,

Thank you for submitting your manuscript to PLOS ONE. After careful consideration, we feel that it has merit but does not fully meet PLOS ONE’s publication criteria as it currently stands. Therefore, we invite you to submit a revised version of the manuscript that addresses the points raised during the review process.

The manuscript has been evaluated by three reviewers, and their comments are available below.

The reviewers have raised a number of concerns that need attention. They request additional information on methodological aspects of the study, revisions to the statistical analyses and they question the conclusions based on the data reported.

Could you please revise the manuscript to carefully address the concerns raised?

We would appreciate receiving your revised manuscript by Mar 19 2020 11:59PM. To enhance the reproducibility of your results, we recommend that if applicable you deposit your laboratory protocols in protocols.io, where a protocol can be assigned its own identifier (DOI) such that it can be cited independently in the future. For instructions see: http://journals.plos.org/plosone/s/submission-guidelines#loc-laboratory-protocols

We look forward to receiving your revised manuscript.

Kind regards,

Nancy Beam, PhD

Staff Editor

PLOS ONE

Journal Requirements:

"A payment for labour of the nurse who participated in the study was covered by NanoPass Technologies Ltd, Israel. Efrat Kochba and Yotam Levin - employees of NanoPass Technologies Ltd participated in preparation of study design and reviewal of the manuscript."         

Please respond by return e-mail so that we can amend your financial disclosure and competing interests on your behalf.

"A payment for labour of the nurse who participated in the study was covered by NanoPass Technologies Ltd, Israel. Efrat Kochba and Yotam Levin - employees of NanoPass Technologies Ltd participated in preparation of study design and reviewal of the manuscript."

We note that you received funding from a commercial source: NanoPass Technologies Ltd

Reviewers' comments:

Reviewer's Responses to Questions

**Comments to the Author**

1. Is the manuscript technically sound, and do the data support the conclusions?

Reviewer #1: No

Reviewer #2: Partly

Reviewer #3: Yes

2. Has the statistical analysis been performed appropriately and rigorously? 

Reviewer #1: No

Reviewer #2: I Don't Know

Reviewer #3: Yes

3. Have the authors made all data underlying the findings in their manuscript fully available?

Reviewer #1: No

Reviewer #2: No

Reviewer #3: Yes

4. Is the manuscript presented in an intelligible fashion and written in standard English?

Reviewer #1: No

Reviewer #2: Yes

Reviewer #3: Yes

5. Review Comments to the Author

Reviewer #1: The authors conducted a (partly) double-blinded three arm placebo-controlled clinical trial to compare the Intradermal injection of lidocaine with a placebo (100μL of saline) as well as doing nothing to provide rapid local anaesthesia for peripheral intravenous cannulation with respect to pain after cannulation. The comparison with lidocaine was done with intraindividual comparisons. They concluded, that the described MicronJet600 technique was safe and effective for providing rapid local anesthesia. The technique is also considered promising for local anesthesia in such procedures as blood gas sampling and minor surgical interventions such as excision of small skin lesions, or surgical sutures removal.

There are major points from the methods point of view which conflict the validity, but may be possibly addressed in a review.

The authors are encouraged to include an individual patient data file which may be anonymized, but let the reviewer and the reader follow the arguments.

Second, I welcome very much the effort to make the presentation consistent with the CONSORT guideline. My impression is, that this is mostly done successful. However, every point of the CONSORT like 3b,6b,7b, 11b, 14b, 17b and 20 should be addressed, at least with the statement like “no changes to the protocol are made”.

The missing line numbers influence the style of my comments:

The general comments are:

1. Restructuring along the CONSORT checklist recommendation is necessary. Some parts of the introduction belong to the Method section (treatments). Most of the study designs section is in-/exclusion criteria (section setting) as well as treatment (section interventions) which does not match to study design. A separate figure explaining the study design is helpful. Please give details / differences between applied treatments to illustrated mantainance to blinding.

2. The conclusion is not supported by the data. A reformulation along the primary hypothesis is necessary.

3. A limitation section is missing in the discussion section.

4. A statement about the ethical consideration, i.e. conduct in a group of healthy volunteers, may be worth to include in the discussion.

5. The number of patients (including diseases) and healthy volunteers should be given.

6. The intraindividual comparisons are confounded with the application side. This effect should by addressed in the limitation section “as unclear effect on the study result” and has implication on the conclusion.

7. The study is only blinded for the group1 comparison. Group2 is at most observer blinded. Effectiveness of “Blinding” is difficult to assess, as the allocation is determined by side and thus only the allocation to the type of the control group might be unclear. This should be made clear; the study design classification should be “open label” instead of blinded. The authors are welcome to describe the efforts to implement blinding.

8. The abstract is not structured according to CONSORT recommendation.

9. Definition of the primary endpoint variable is missing in the abstract as well as the method section. Please give the time of evaluation when defining the primary endpoint variable as well.

10. Characterization of secondary variables is missing.

11. At the end of the “Introduction” a clearly specified primary hypotheses (which can be answered by a suitable statistical test) is missing.

12. The text following “The results of this ..” should be deleted, while no data are presented or published.

13. Give the location of the clinic in the setting section.

14. Give the initials of the observers.

15. Please give explanation, what is gained from the second control group. Although the saline group can be followed the second control group is difficult to understand. Note, the comparison for the MicronJet600 technique (control group1 vs control group2) is not indicated as a primary research question.

16. The comparison within group2 (lido vs no pretreatment) does not answer the question about MicronJet600 use, as the treatment is confounded with the MicronJet600 use.

17. The statistical analysis model may be inadequate, depending on the primary research question/hypothesis. In particular with the comparisons to the lido pretreatment more data are available, than used for the particular analysis. A suitable linear mixed model might be helpful to address intra-patient variability as well as all treatments. Linear contrast help to answer specific questions while protecting for type one error inflation.

18. Details of the repeated measures ANOVA model should be given. I do not see two between factors here. Linear mixed effects model with a clear statement about within factors, random factors and covariance structure assumption is necessary (see comment 17).

19. Sample size calculation is difficult to understand, which the primary endpoint is not referred to, a reference of the numbers is not given and the clinical relevance of the effects is not clear, the link to a hypothesis reflecting the design is not mentioned.

20. P-Values should be given by at least 3 decimal digits.

21. Classification of VAS scale seem to be arbitrary.

Reviewer #2: Thank you for the opportunity to review your manuscript. This study concerns patient’s perception of IV cannulations and evaluates a device for ensuring immediate pain relief as an alternative to topically applied local anesthetics that requires a pause in the procedure. The design is described as a randomized double-blinded placebo controlled clinical trial and according to the study synopsis the study also follows GCP guidelines.

However, the study, as described both in researchregistry.com, the study synopsis and in the manuscript, has substantial flaws in its design. The randomization has not been implemented to optimize the study trustworthiness and the blinding procedures are insufficient.

Introduction:

How do you substantiate the claim that MicronJet600 is “the foremost device” of the commercially available?

The last paragraph would benefit by removing the comparison with the previous device and instead focus on presenting the primary and secondary objectives of the study, preferentially as testable hypothesis.

Methods

Who prepared the MicroJet600 devices before use, i.e. added saline or 2% lidocaine? Observer 1? Was it the same research nurse that inserted the 18G needles? Why was not the right vs left arm also randomized to avoid potential systematic technical issues with the IV cannulations and also blind the observers? Was the the order between right and left arm cannulations identical between all subjects?

The randomization appears to be into groups that received lidocaine (left arm) – saline (right arm), or lidocaine (left arm) – no injection (right arm). Thus, according to the protocol that was made public before the first patient was included, both the research nurse and observers potentially knew which arm that in all cases had been injected with lidocaine. This is hardly a blinded protocol. Furthermore, from the patient’s point of view the first group might have been considered blinded provided that the protocol had stipulated a randomization of which arm that received saline or lidocaine. But this was not the case according to the protocol.

Statistics

How did you test the distribution of the data, i.e. whether parametric tests are valid and mean +-SD is a valid descriptor?

Results

The results section can be substantially shorter and would benefit from being presented in line with the hypotheses that are tested.

Table 1: typo with a , instead of a .

Protocol considerations

The procedure for blinding of the patient and of the observers are flawed. The protocol that was made public before the start of the inclusion of patients states that all injections with active substance are made in the left arm, and placebo or no injection are always done in the right arm. Also Observer 1, who prepared the devices for injection, are involved in evaluations of the patients.

The amount of pain inflicted on the patients by the injection of placebo or active substance, compared to the pain inflicted by the insertion of the 18 G needle would by itself have been an interesting comparison.

Discussion

Please remove unpublished data from discussion. It does not substantiate the publication. Again, please state aims and hypothesis in introduction.

Conclusion

Should be directly in line with the stated objectives. Please avoid other assumptions in the conclusion.

Financial disclosure

The text in the manuscript does not match the text in the additional information.

Data availability

The datapoints has not been made available in this version. Please see the last sentence:

”The PLOS Data policy requires authors to make all data underlying the findings described in their manuscript fully available without restriction, with rare exception (please refer to the Data Availability Statement in the manuscript PDF file). The data should be provided as part of the manuscript or its supporting information, or deposited to a public repository. For example, in addition to summary statistics, the data points behind means, medians and variance measures should be available.”

Reviewer #3: Intradermal injection of lidocaine with a microneedle device to provide rapid local anaesthesia for peripheral intravenous cannulation: A randomised double-blind placebo-controlled clinical trial

Nice study, very Practical among the pediatric population and can be used in adult as well.

This study is done well with good statistical analysis.

In our institute we start using the J-tip and you can check it on www.jtip.com

It's Needle-free injecting System for subcutaneous administration of Novocaine

We found some benefit from using Compared to the EMLA cream

In the mean while we are generating a study very close to your study.

6. PLOS authors have the option to publish the peer review history of their article (what does this mean?). If published, this will include your full peer review and any attached files.

Reviewer #1: No

Reviewer #2: No

Reviewer #3: Yes: John Seif MD, MBA

---

## [Author Response · Author response to Decision Letter 0]

6 Aug 2020

4 August 2020

Dear Reviewers, 

Please consider a revised version of our manuscript titled " Intradermal injection of lidocaine with a microneedle device to provide rapid local anesthesia for peripheral intravenous cannulation: A randomized double-blind placebo-controlled clinical trial”. The manuscript was revised extensively, we greatly apologize for the delay. The response to the reviewers' comments is in the "Response to Reviewers" file.

Yours sincerely, 

Dr Alexey Rzhevskiy 

Department of Clinical Medicine, Faculty of Medicine and Health Sciences, 

Macquarie University, Sydney, Australia 

Institute of Molecular Medicine, Sechenov First Moscow State Medical 

University, Moscow, Russia

---

## [Decision Letter · Decision Letter 1]

18 Jan 2021

PONE-D-19-27010R1

Intradermal injection of lidocaine with a microneedle device to provide rapid local anaesthesia for peripheral intravenous cannulation: A randomised double-blind placebo-controlled clinical trial

PLOS ONE

Dear Dr. Rzhevskiy,

Thank you for submitting your manuscript to PLOS ONE. After careful consideration, we feel that it has merit but does not fully meet PLOS ONE’s publication criteria as it currently stands. Therefore, we invite you to submit a revised version of the manuscript that addresses the points raised during the review process.

The manuscript has been re-evaluated by three reviewers, and their comments are available below. You will see the reviewers have commented on the thoroughness of the response to previous reviewer comments. However, the reviewers have also highlighted persisting critical concerns and the manuscript will need significant revision before it can be considered for publication – you should anticipate that the reviewers will be re-invited to assess the revised manuscript, so please ensure that your revision is thorough. I have outlined some of the key concerns noted by the reviewers below, but you should respond all concerns mentioned by the reviewers in your response-to-reviewers document. 

The key concerns noted by the reviewers relate to the blinding procedure, the lack of comparison of the study intervention to a normal intradermal needle, and the sample size calculation. Additionally, the reviewers requested clarification regarding the randomization procedure. These issues have limitations for the interpretation of the results and should be explored.

We look forward to receiving your revised manuscript.

Kind regards,

Danielle Poole

Staff Editor

PLOS ONE

Reviewers' comments:

Reviewer's Responses to Questions

**Comments to the Author**

1. If the authors have adequately addressed your comments raised in a previous round of review and you feel that this manuscript is now acceptable for publication, you may indicate that here to bypass the “Comments to the Author” section, enter your conflict of interest statement in the “Confidential to Editor” section, and submit your "Accept" recommendation.

Reviewer #2: (No Response)

Reviewer #4: (No Response)

Reviewer #5: (No Response)

2. Is the manuscript technically sound, and do the data support the conclusions?

Reviewer #2: No

Reviewer #4: Partly

Reviewer #5: Partly

3. Has the statistical analysis been performed appropriately and rigorously? 

Reviewer #2: Yes

Reviewer #4: Yes

Reviewer #5: Yes

4. Have the authors made all data underlying the findings in their manuscript fully available?

Reviewer #2: Yes

Reviewer #4: Yes

Reviewer #5: (No Response)

5. Is the manuscript presented in an intelligible fashion and written in standard English?

Reviewer #2: Yes

Reviewer #4: Yes

Reviewer #5: No

6. Review Comments to the Author

Reviewer #2: Thank you for addressing the comments and vastly improving the manuscript. Now it is easy to follow your design and experimental setup.

A major issue remains. In this study, all patients according to the protocol received the active substance in the right arm and the left arm was always control. The randomization was to either receive saline injected in the left arm or not to receive an injection of saline. This was known to the investigators that made the measurements.

The protocol stating this procedure, the active substance in the right arm and saline/no-treatment in the left arm, was published on the web. Thus it cannot be ruled out that also the test subject knew which side that received the active substance. Accordingly, the study can not be said to be blinded. This needs to be changed throughout the title, abstract and manuscript.

The conclusion should be based on data from this study and answer the aim of the study . Thus the first sentence in the conclusion in the abstract is relevant but not the second sentence. I suggest that you delete it.

The conclusion in the main text is currently a repetition of results and also a sales pitch for more uses of the device. It needs to be revised.

Reviewer #4: The authors performed a double-blinded randomized trial to assess the efficacy and safey of a microneedling device to provide rapid anesthesia for the insertion of a peripheral intravenous catheter. 100 health volunteers were randomized into two groups: one with microneedle with lido 2% and saline in contralateral arm, second group one with microneedle with lido 2% and no injection in the contralateral arm. The subjects served as there own control. An 18G cannula was inserted in both arms. Intradermal injection with lido 2% reduced pain significantly and subjects preferred lido. It was concluded that injection with microjet600 was safe and effective option for rapid local anesthesia for peripheral iv cannulation.

Control group that compared a to a regular intradermal injection is not included. So it is not clear whether the injection with the micronjet600 is necessary to achieve the effect.

Would a normal intradermal needle would provide the same result?

Was the person that applied the lido 2%/saline with the micronjet blinded?

P6, sample size � calculation of the sample size is unclear. Wat is meant by effect size 1 with and SD of 2.2? Is one point on the VAS score meant here?

P9, adverse events � there was not powered for adverse events, question arises whether safety can be really assessed based on these data. However, it is not expected that intradermal injections cannot be considered safe, as local anesthesia with lidocaine is common practice.

P9, first paragraph of discussion: the results do not have to be repeated. Suffices to state main findings here.

Limitations:

No control with normal needle. The notion that the use of MicronJet600 is speculative

Reviewer #5: The authors have responded to most of the issues that Reviewer 1 listed about the statistical aspects of the paper. There are a few remaining issues:

1. The randomization procedure: complete, permuted blocks, etc. (see Rosenberger and Lachin, Randomization in Clinical Trials, 2016, Wiley for details). Excel is NOT a randomization procedure.

2. Sample size is still unclear. What is a treatment effect of 1? Does this mean a change in mean VAS of 1? Be more specific. Where is the standard deviation of 2.2 from? You need to reference a pilot study or the literature.

3. There are voluminous missing and incorrect definite and indefinite articles.

7. PLOS authors have the option to publish the peer review history of their article (what does this mean?). If published, this will include your full peer review and any attached files.

Reviewer #2: No

Reviewer #4: **Yes: **R. Arthur Bouwman

Reviewer #5: No

---

## [Author Response · Author response to Decision Letter 1]

4 Mar 2021

Dear Reviewers,

Please see the attached docx document "Response to Reviewers2,27.02.2021". You can also see the text from this document below.

Reviewer #2: Thank you for addressing the comments and vastly improving the manuscript. Now it is easy to follow your design and experimental setup.

A major issue remains. In this study, all patients according to the protocol received the active substance in the right arm and the left arm was always control. The randomization was to either receive saline injected in the left arm or not to receive an injection of saline. This was known to the investigators that made the measurements.

The protocol stating this procedure, the active substance in the right arm and saline/no-treatment in the left arm, was published on the web. Thus it cannot be ruled out that also the test subject knew which side that received the active substance. Accordingly, the study can not be said to be blinded. This needs to be changed throughout the title, abstract and manuscript.

Response: Yes, the blindness could has been performed more rigorously. In the title, abstract and manuscript “double-blinded” was changed to “open-label”. 

The conclusion should be based on data from this study and answer the aim of the study. Thus the first sentence in the conclusion in the abstract is relevant but not the second sentence. I suggest that you delete it.

Response: Done.

The conclusion in the main text is currently a repetition of results and also a sales pitch for more uses of the device. It needs to be revised.

Response: An excess information regarding the results of the study was removed from the “Conclusions” section. However, we can not entirely rewrite the conclusions because conclusions are always drawn from results. 

Reviewer #4: The authors performed a double-blinded randomized trial to assess the efficacy and safey of a microneedling device to provide rapid anesthesia for the insertion of a peripheral intravenous catheter. 100 health volunteers were randomized into two groups: one with microneedle with lido 2% and saline in contralateral arm, second group one with microneedle with lido 2% and no injection in the contralateral arm. The subjects served as there own control. An 18G cannula was inserted in both arms. Intradermal injection with lido 2% reduced pain significantly and subjects preferred lido. It was concluded that injection with microjet600 was safe and effective option for rapid local anesthesia for peripheral iv cannulation.

Control group that compared a to a regular intradermal injection is not included. So it is not clear whether the injection with the micronjet600 is necessary to achieve the effect.

Would a normal intradermal needle would provide the same result?

Response: Lidocaine injection with a normal hypodermic needle would probably have similar efficacy in terms of anaesthesia. However, because injection with a normal hypodermic needle is painful and highly-invasive, it is commonly not used in clinical practice prior to intravenous cannulation. Thus, minimally invasive and none-invasive approaches for intradermal delivery of anaesthetics have been actively investigated, please see a systematic review by Bond M, Crathorne L, Peters J, et al. 2015 (the first reference in the article). In most of the clinical trials investigating such minimally invasive or none-invasive approaches, control group with a regular intradermal injection was not included. As an example of such studies, please see Spanos S, Booth R, Koenig H, et al. 2008 (reference 7 in the article) and Lysakowski C, Dumont L, Tramèr MR, Tassonyi E 2003 (reference 23 in the article).

Was the person that applied the lido 2%/saline with the micronjet blinded?

Response: Yes, the nurse who applied the lido 2%/saline with the MicronJet600 was blinded.

P6, sample size � calculation of the sample size is unclear. Wat is meant by effect size 1 with and SD of 2.2? Is one point on the VAS score meant here?

Response: The following has been added to the text of the article in the “Sample Size” section, marked with yellow:

“A sample size of 40 subjects per group was calculated to detect an effect size (expected difference on the VAS score between two time points at the specific distance) 1 with standard deviation of the effect at 2.2 using a paired t-test with 80% power and 5% type I error rate assuming two-sided significance testing procedure. At the same time, additional 22 subjects (102 subjects in total) were enrolled in order to overcome dropout scores.”

P9, adverse events � there was not powered for adverse events, question arises whether safety can be really assessed based on these data. However, it is not expected that intradermal injections cannot be considered safe, as local anesthesia with lidocaine is common practice.

Response: The limitation has been added to the text of the article.

P9, first paragraph of discussion: the results do not have to be repeated. Suffices to state main findings here.

Response: The information on the results, presented in the first paragraph of the “Discussion” section, is coherent with the discussion of these results. We greatly apologize but in our opinion, this information can not be just simply removed without compromising the overall meaning of the discussion.

Limitations:

No control with normal needle. The notion that the use of MicronJet600 is speculative

Response: We agree that the study was not powered for adverse events, we did not perform statistical testing procedures on these data. The limitation has been added to the text of the article.

Reviewer #5: The authors have responded to most of the issues that Reviewer 1 listed about the statistical aspects of the paper. There are a few remaining issues:

1. The randomization procedure: complete, permuted blocks, etc. (see Rosenberger and Lachin, Randomization in Clinical Trials, 2016, Wiley for details). Excel is NOT a randomization procedure.

Response: The following has been added to the text of the article in the “Participants” section:

Response: Simple randomization was preformed to allocate subjects into 2 groups using Microsoft Excel random number generator.

2. Sample size is still unclear. What is a treatment effect of 1? Does this mean a change in mean VAS of 1? Be more specific. Where is the standard deviation of 2.2 from? You need to reference a pilot study or the literature.

Response: The following has been added to the text of the article in the Sample Size section, marked with yellow:

“A sample size of 40 subjects per group was calculated to detect an effect size (expected difference on the VAS score between two time points at the specific distance) 1 with standard deviation of the effect at 2.2 using a paired t-test with 80% power and 5% type I error rate assuming two-sided significance testing procedure. At the same time, additional 22 subjects (102 subjects in total) were enrolled in order to overcome dropout scores.”

3. There are voluminous missing and incorrect definite and indefinite articles.

Response: The manuscript has been revised and necessary corrections have been made. However, we would be happy if the Reviewer #5 specify the incorrect definite and indefinite articles.

---

## [Decision Letter · Decision Letter 2]

13 Apr 2021

PONE-D-19-27010R2

Intradermal injection of lidocaine with a microneedle device to provide rapid local anaesthesia for peripheral intravenous cannulation: A randomised open-label placebo-controlled clinical trial

PLOS ONE

Dear Dr. Rzhevskiy,

Thank you for submitting your manuscript to PLOS ONE. After careful consideration, we feel that it has merit but does not fully meet PLOS ONE’s publication criteria as it currently stands. Therefore, we invite you to submit a revised version of the manuscript that addresses the points raised during the review process.

The manuscript has been re-evaluated by three reviewers, and their comments are available below. You will see the reviewers have commented on the strengths of this revision. However, they have also raised a number of minor concerns that should be addressed before the manuscript can be further considered for publication.

The key concerns noted by the reviewers relate to the overall reporting in the manuscript. Specifically, the reviewers have requested that your manuscript be reviewed by an English language service to improve clarity.  

We look forward to receiving your revised manuscript.

Kind regards,

Danielle Poole

Staff Editor

PLOS ONE

Journal Requirements:

Reviewers' comments:

Reviewer's Responses to Questions

**Comments to the Author**

1. If the authors have adequately addressed your comments raised in a previous round of review and you feel that this manuscript is now acceptable for publication, you may indicate that here to bypass the “Comments to the Author” section, enter your conflict of interest statement in the “Confidential to Editor” section, and submit your "Accept" recommendation.

Reviewer #2: (No Response)

Reviewer #4: All comments have been addressed

Reviewer #5: All comments have been addressed

2. Is the manuscript technically sound, and do the data support the conclusions?

Reviewer #2: Partly

Reviewer #4: Yes

Reviewer #5: (No Response)

3. Has the statistical analysis been performed appropriately and rigorously? 

Reviewer #2: Yes

Reviewer #4: Yes

Reviewer #5: (No Response)

4. Have the authors made all data underlying the findings in their manuscript fully available?

Reviewer #2: Yes

Reviewer #4: Yes

Reviewer #5: (No Response)

5. Is the manuscript presented in an intelligible fashion and written in standard English?

Reviewer #2: No

Reviewer #4: Yes

Reviewer #5: No

6. Review Comments to the Author

Reviewer #2: Thank you for the adjustments to the manuscript. It addresses most of my concerns.

However, the present version of the submission still have issues that needs to be addressed.

Please update the abstract in the submission system to be identical with the abstract in the manuscript.

The study cited on page 3 was registered 2007 and has not yet been published. I would suggest that you remove the last sentence in the paragraph completely as the claimed support for the results cannot be verified.

The language needs to be edited by English editing service.

Examples:

"Consequently, the average pain scores for cannulation with no pretreatment and cannulation with the preliminary intradermal injection of the placebo were comparable in both studies – the current study and the study by Lysakowski et al.". Please rewrite without the -

"Thus, the average 100-poin VAS pain scores versus 10-poin NVS were: 39.7 and 3.9, and 41.5 and 4.2 for cannulation". Should be point instead of poin.

"It is anticipated that the technique is appropriate for a middle-term small surgical procedures such as excision of small skin lesions, or surgical skin staples or sutures removal [24], nevus removal. " Should be rephrased into correct english.

Table 1

An accidential , instead of .

Citing of references needs to be consistent and according to PlsOne standards. Eg, in the same paragraph the same paper is referenced differently:

...in comparison with the study by Lysakowski, Dumont, Tramer and Tasonyi. (500 μL) [23]

- The last author misspelled

...the study by Lysakowski et al. [23]

Including own unpublished data as in the last paragraph in the discussion (page 10) is still not appropriate as the results cannot be verified against methods used, nor have they been peer-reviewed. If the results holds for dissemination they should be published within in the present or in a separate study.

Conclusion

I expect the conclusion to be brief and directly related to the aims of the study. It would benefit by being substantially shorter and not be a repetition of the results.

Reviewer #4: This is the revised version of the manuscript “Intradermal injection of lidocaine with a microneedle device to provide rapid local anaesthesia for peripheral intravenous cannulation: A randomised open-label placebocontrolled clinical trial” by Rzhevskiy and co-authors.

The manuscript has improved and only minor issues remain.

Discussion

The first paragraph of the discussion should state the main findings, a short summary, without repeating the results section.

Figures

Resolution of the figure 2 and 3 are not optima land should be improved

Figure 2

Layout figure 2 is not optimal and can be improved

Figure 5

Is figure correct? In this figure the total of the bars per group should be 100%, this seems to be the case in the no pretreatment group and placebo group, but not in the 2% lido group: when the total of the bars is more than 100%

Reviewer #5: (No Response)

7. PLOS authors have the option to publish the peer review history of their article (what does this mean?). If published, this will include your full peer review and any attached files.

Reviewer #2: No

Reviewer #4: No

Reviewer #5: No

---

## [Author Response · Author response to Decision Letter 2]

8 May 2021

Reviewer #2: Thank you for the adjustments to the manuscript. It addresses most of my concerns.

However, the present version of the submission still have issues that needs to be addressed.

Comment: Please update the abstract in the submission system to be identical with the abstract in the manuscript.

Response: Done

Comment: The study cited on page 3 was registered 2007 and has not yet been published. I would suggest that you remove the last sentence in the paragraph completely as the claimed support for the results cannot be verified.

Response: Done

Comment: The language needs to be edited by English editing service.

Response: Edited

Examples:

Comment: "Consequently, the average pain scores for cannulation with no pretreatment and cannulation with the preliminary intradermal injection of the placebo were comparable in both studies – the current study and the study by Lysakowski et al.". Please rewrite without the –

Response: Done

Comment: "Thus, the average 100-poin VAS pain scores versus 10-poin NVS were: 39.7 and 3.9, and 41.5 and 4.2 for cannulation". Should be point instead of poin.

Response: Done

Comment: "It is anticipated that the technique is appropriate for a middle-term small surgical procedures such as excision of small skin lesions, or surgical skin staples or sutures removal [24], nevus removal. " Should be rephrased into correct english.

Response: Done

Comment: Table 1

An accidential , instead of .

Response: Corrected

Comment: Citing of references needs to be consistent and according to PlsOne standards. Eg, in the same paragraph the same paper is referenced differently:

...in comparison with the study by Lysakowski, Dumont, Tramer and Tasonyi. (500 μL) [23]

- The last author misspelled

...the study by Lysakowski et al. [23]

Response: Done

Comment: Including own unpublished data as in the last paragraph in the discussion (page 10) is still not appropriate as the results cannot be verified against methods used, nor have they been peer-reviewed. If the results holds for dissemination they should be published within in the present or in a separate study.

Response: The paragraph has been removed

Comment: Conclusion

I expect the conclusion to be brief and directly related to the aims of the study. It would benefit by being substantially shorter and not be a repetition of the results.

Response: Done

Reviewer #4: This is the revised version of the manuscript “Intradermal injection of lidocaine with a microneedle device to provide rapid local anaesthesia for peripheral intravenous cannulation: A randomised open-label placebocontrolled clinical trial” by Rzhevskiy and co-authors.

The manuscript has improved and only minor issues remain.

Comment: Discussion

The first paragraph of the discussion should state the main findings, a short summary, without repeating the results section.

Response: Done

Comment: Figures

Resolution of the figure 2 and 3 are not optima land should be improved

Response: Improved

Comment: Figure 2

Layout figure 2 is not optimal and can be improved

Response: Improved

Comment: Figure 5

Is figure correct? In this figure the total of the bars per group should be 100%, this seems to be the case in the no pretreatment group and placebo group, but not in the 2% lido group: when the total of the bars is more than 100%

Response: Yes, the figure is correct. In this figure, the total of the bars per group should not be 100%. For better understanding, such category as VAS-score ⩽ 20 includes all cases from 0 to 20, therefore this category at the same time involves such categories as VAS-score ⩽ 10 and VAS-score=0.

Reviewer #5: (No Response)

---

## [Decision Letter · Decision Letter 3]

9 Dec 2021

Intradermal injection of lidocaine with a microneedle device to provide rapid local anaesthesia for peripheral intravenous cannulation: A randomised open-label placebo-controlled clinical trial

PONE-D-19-27010R3

Dear Dr. Alexey Rzhevskiy

We’re pleased to inform you that your manuscript has been judged scientifically suitable for publication and will be formally accepted for publication once it meets all outstanding technical requirements.

Kind regards,

Ehab Farag, MD FRCA FASA

Academic Editor

PLOS ONE

Additional Editor Comments (optional):

Reviewers' comments:

Reviewer's Responses to Questions

**Comments to the Author**

1. If the authors have adequately addressed your comments raised in a previous round of review and you feel that this manuscript is now acceptable for publication, you may indicate that here to bypass the “Comments to the Author” section, enter your conflict of interest statement in the “Confidential to Editor” section, and submit your "Accept" recommendation.

Reviewer #2: All comments have been addressed

Reviewer #4: All comments have been addressed

Reviewer #5: All comments have been addressed

2. Is the manuscript technically sound, and do the data support the conclusions?

Reviewer #2: Yes

Reviewer #4: Yes

Reviewer #5: (No Response)

3. Has the statistical analysis been performed appropriately and rigorously? 

Reviewer #2: Yes

Reviewer #4: Yes

Reviewer #5: (No Response)

4. Have the authors made all data underlying the findings in their manuscript fully available?

Reviewer #2: Yes

Reviewer #4: Yes

Reviewer #5: (No Response)

5. Is the manuscript presented in an intelligible fashion and written in standard English?

Reviewer #2: Yes

Reviewer #4: Yes

Reviewer #5: (No Response)

6. Review Comments to the Author

Reviewer #2: No further issues. Thanks for the opportunity to review tjis manuscript.

Reviewer #4: This is the third revision of the manuscript. The authors improved the manuscript. no further comments remain

Reviewer #5: (No Response)

7. PLOS authors have the option to publish the peer review history of their article (what does this mean?). If published, this will include your full peer review and any attached files.

Reviewer #2: No

Reviewer #4: No

Reviewer #5: No

---

## [Editor Report · Acceptance letter]

21 Jan 2022

PONE-D-19-27010R3 

Intradermal injection of lidocaine with a microneedle device to provide rapid local anaesthesia for peripheral intravenous cannulation: A randomised open-label placebo-controlled clinical trial 

Dear Dr. Rzhevskiy:

I'm pleased to inform you that your manuscript has been deemed suitable for publication in PLOS ONE. Congratulations! Your manuscript is now with our production department. 

Kind regards, 

on behalf of

Dr. Ehab Farag 

Academic Editor

PLOS ONE